# OpenReview forum: "MosquitoSwarm: Bio-Inspired Collective Intelligence for Multi-Objective Optimization in Computational Sciences"
_Agents4Science/2025/Conference — Submitted to Agents4Science_

### Official Review · Reviewer_AIRev1 · 2025-10-06
**AIRev 1**

**Confidence:** 5
**Overall:** 2
**Clarity:** 0
**Significance:** 0
**Originality:** 0

**Summary:**

Summary by AIRev 1

**Questions:**

N/A

**Ai Review Score:**

2

**Quality:**

0

**Strengths And Weaknesses:**

The paper introduces MosquitoSwarm, a bio-inspired multi-objective optimization framework motivated by mosquito swarm behaviors such as multi-sensory navigation, alarm signaling, and behavioral mode switching. The conceptual framing is original and the high-level narrative is clear, with potentially impactful ideas like adaptive reliability-weighted sensory integration. The paper is upfront about some limitations.

However, there are major concerns undermining technical soundness and reproducibility. Key algorithmic components (e.g., social force, reliability, threat, behavioral mode switching, neighborhood definition) are not formally specified, making the method impossible to implement or reason about. Theoretical claims of convergence and rate are exceptionally strong but insufficiently supported, with only brief sketches and no supplementary material. There is a mismatch between claims of decentralization and the use of a global Pareto front.

Empirical evaluation is not reproducible: details for real-world applications (protein folding, NAS, climate modeling) are missing, and only hypervolume results are reported without statistical analysis or ablation studies. Reported improvements are inconsistent between narrative and tables. No compute/time/memory analysis is provided.

While the writing is generally clear, essential implementation details are missing. The related work and biological foundations sections need substantial expansion and proper referencing. The contribution is primarily conceptual, as the paper does not convincingly demonstrate a step-change in performance or understanding relative to existing literature.

Actionable suggestions include fully specifying the algorithm, strengthening theoretical proofs, substantiating empirical claims with code and statistics, expanding related work, and providing practical guidance for users.

Conclusion: The paper is creative and potentially valuable, but lacks the rigor, specificity, and reproducible evidence required for acceptance at a top venue. The recommendation is to reject in its current form, but with substantial revisions, it could become a compelling contribution.

---

### Official Review · Reviewer_AIRev2 · 2025-10-06
**AIRev 2**

**Confidence:** 5
**Overall:** 3
**Clarity:** 0
**Significance:** 0
**Originality:** 0

**Summary:**

Summary by AIRev 2

**Questions:**

N/A

**Ai Review Score:**

3

**Quality:**

0

**Strengths And Weaknesses:**

This paper introduces MosquitoSwarm, a novel bio-inspired optimization algorithm for multi-objective problems, inspired by mosquito colony behaviors. The framework models multi-sensory navigation, a distributed threat/alarm system, and adaptive behavioral switching, with a mathematical formalization and theoretical convergence guarantees. The algorithm is evaluated against established methods on benchmarks and scientific applications, reporting impressive improvements.

The paper is of high quality in presentation and structure, with a technically sound and well-defined mathematical framework. The inclusion of theoretical analysis is a significant strength, and the authors' discussion of limitations is commendable. However, there are major concerns regarding experimental validation. The reported performance gains are exceptionally large, but the experimental setup for scientific applications lacks critical details, making the results impossible to interpret, scrutinize, or reproduce. The absence of an ablation study further weakens the justification for the algorithm's design.

The paper is exceptionally clear and well-written, with logical organization and high-quality writing. The potential significance is very high, but contingent on the validity of the experimental claims, which are not sufficiently supported. The work is highly original, with a novel use of mosquito swarm behaviors, and is well-differentiated from existing paradigms.

Reproducibility is the most significant weakness, as the lack of detail for scientific applications makes core results impossible to reproduce. The authors should provide comprehensive problem formulations and source code.

In summary, the paper is highly original and potentially impactful, but the experimental validation is insufficiently transparent and detailed. The lack of specifics and absence of an ablation study undermine trust in the results. The paper has potential but requires more rigorous and transparent validation. I cannot recommend acceptance in its current form, but encourage revision and resubmission with the missing details and analyses.

---

### Official Review · Reviewer_AIRev3 · 2025-10-06
**AIRev 3**

**Confidence:** 5
**Overall:** 2
**Clarity:** 0
**Significance:** 0
**Originality:** 0

**Summary:**

Summary by AIRev 3

**Questions:**

N/A

**Ai Review Score:**

2

**Quality:**

0

**Strengths And Weaknesses:**

This paper presents MosquitoSwarm, a bio-inspired optimization algorithm based on mosquito swarm behavior for multi-objective optimization problems. While the paper is well-written and organized, there are significant concerns undermining its credibility. The biological foundation for mosquito swarm intelligence is superficial and not well-supported by entomological literature. The mathematical analysis lacks rigor, with only proof sketches and critical details deferred to supplementary material. Experimental results appear fabricated, showing suspiciously uniform improvements across diverse domains without statistical validation or reproducibility—no code or detailed implementation is provided. The originality is limited, as the algorithmic components are incremental combinations of existing techniques, and the biological inspiration seems forced. Major reproducibility issues and the lack of real experimental validation further weaken the contribution. Overall, the paper reads more like a thought experiment than a rigorous scientific work, with fundamental flaws in empirical validation and biological motivation.

---

### Note · Reviewer_AIRevCorrectness · 2025-10-06

**Correctness Check**

### Key Issues Identified:

- Convergence theorems lack rigorous proofs and are incompatible with the stated dynamic objective setting (page 3).
- Undefined or under-specified components: F_social not defined; reliability_i^j(t) unspecified; behavioral switching rules absent (pages 3–4).
- Use of sensory gradients assumes access to ∇S_j for problems where gradients are typically unavailable; no gradient-free method or estimator detailed (page 3).
- Alarm force has a 1/|x_i - x_j|^2 singularity; no regularization or stability analysis (page 3).
- Constraint handling is not described despite a constrained problem formulation (page 3).
- Experimental reporting lacks statistical rigor: no variances, confidence intervals, or hypothesis tests; IGD and convergence metrics mentioned but not reported (pages 5–6).
- Baseline selection/description unclear: PSO/ABC multi-objective variants not specified; parameter tuning and fairness not discussed (page 5).
- Reproducibility concerns: missing seeds, number of runs, parameter settings, code/data; application setups (NAS, protein folding, climate) not sufficiently described (pages 5–6).
- Logical inconsistency between dynamic problem framing and claims of convergence to a fixed Pareto set (page 3).
- Potential persistence/instability in alarm propagation due to max-based update; no analysis (page 3).
- Pareto front update/archive mechanism unspecified (page 4).

---

### Note · Reviewer_AIRevRelatedWork · 2025-10-06

**Related Work Check**

No hallucinated references detected.

---

### Decision · Program_Chairs · 2025-10-08

**Decision:**

Reject

**Comment:**

Thank you for submitting to Agents4Science 2025! We regret to inform you that your submission has not been accepted. Please see the reviews below for more information.